# A sex difference in the response of the rodent postsynaptic density to synGAP haploinsufficiency

Tara L Mastro[1]*, Anthony Preza[1], Shinjini Basu[2], Sumantra Chattarji[2,3], Sally M Till[2], Peter C Kind[2,3], Mary B Kennedy[1]*

[1]Division of Biology and Biological Engineering, Caltech, Pasadena, United States; [2]Simons Initiative for the Developing Brain, Centre for Discovery Brain Sciences, University of Edinburgh, Edinburgh, United Kingdom; [3]Centre for Brain Development and Repair, Bangalore, India

**Abstract** SynGAP is a postsynaptic density (PSD) protein that binds to PDZ domains of the scaffold protein PSD-95. We previously reported that heterozygous deletion of *Syngap1* in mice is correlated with increased steady-state levels of other key PSD proteins that bind PSD-95, although the level of PSD-95 remains constant (Walkup et al., 2016). For example, the ratio to PSD-95 of Transmembrane AMPA-Receptor-associated Proteins (TARPs), which mediate binding of AMPA-type glutamate receptors to PSD-95, was increased in young *Syngap1*[+/-]mice. Here we show that only females and not males show a highly significant correlation between an increase in TARP and a decrease in synGAP in the PSDs of *Syngap1*[+/-]rodents. The data reveal a sex difference in the adaptation of the PSD scaffold to synGAP haploinsufficiency.

## Introduction

SynGAP is a Ras/Rap GTPase Activating Protein that is specifically expressed in neurons and is highly concentrated in the postsynaptic density (PSD) of glutamatergic synapses in the brain (*Chen et al., 1998*; *Kim et al., 1998*). Mutations that cause heterozygous deletion or dysfunction of the human gene *Syngap1* cause a severe form of intellectual disability (synGAP haploinsufficiency, also called Mental Retardation type 5 [MRD5]) often accompanied by autism and/or seizures (*Berryer et al., 2013*; *Hamdan et al., 2011*; *Hamdan et al., 2009*). In mice, heterozygous deletion of the gene *Syngap1* causes similar neurological deficits; homozygous deletion causes death a few days after birth (*Komiyama et al., 2002*; *Vazquez et al., 2004*).

One function of synGAP is to regulate the balance of active Ras and Rap at the postsynaptic membrane (*Walkup et al., 2015*), thereby controlling the balance of exocytosis and endocytosis of AMPA-type glutamate receptors (*Zhu et al., 2002*) and contributing to regulation of the actin cytoskeleton (*Tolias et al., 2005*). In a recent paper in eLife (*Walkup et al., 2016*), we postulated that synGAP also helps to regulate anchoring of AMPA-type glutamate receptors (AMPARs) in the PSD. AMPARs are tethered to the scaffold protein PSD-95 by auxiliary subunits called TARPs (Transmembrane AMPA Receptor-associated Proteins, *Tomita et al., 2003*). TARPs contain a PDZ ligand that binds to PDZ domains in PSD-95. An early event in induction of long-term potentiation (LTP) is increased trapping of AMPARs that is mediated by enhanced binding of TARPs to PDZ domains (*Opazo and Choquet, 2011*; *Tomita et al., 2005*). SynGAP is also anchored in the PSD by binding of its α1 splice variant to the PDZ domains of PSD-95 (*Kim et al., 1998*; *McMahon et al., 2012*; *Walkup et al., 2016*). SynGAP is nearly as abundant in the PSD fraction as PSD-95, which suggests that it occupies a large fraction of the PDZ domains and can compete with TARPs for binding to PSD-95 (*Chen et al., 1998*; *Dosemeci et al., 2007*). During induction of LTP, calcium/calmodulin-

*For correspondence:
tmastro@caltech.edu (TLM);
kennedym@its.caltech.edu (MBK)

Competing interests: The authors declare that no competing interests exist.

dependent protein kinase II (CaMKII) phosphorylates synGAP, increasing the rate of inactivation of Rap relative to Ras, and, at the same time, causing a decrease in the affinity of synGAP-α1 for the PDZ domains of PSD-95 (*Walkup et al., 2015*; *Walkup et al., 2016*). We postulated that the decreased affinity of synGAP for PSD-95 might contribute to induction of LTP by allowing TARPs and their associated AMPARs to compete more effectively for binding to the PDZ domains and thus increase their anchoring in the PSD. If this hypothesis is correct, one consequence could be that induction of LTP would be disrupted in synGAP heterozygotes because the transient shift in competition for PDZ binding by synGAP would be less potent because of loss of a copy of S*yngap1*. A second possible consequence could be that the steady state level of TARPs bound to PSD-95 in PSDs would be increased in synGAP heterozygotes because the steady state level of synGAP is reduced.

In the study that prompted the present work (*Walkup et al., 2016*), we measured the ratios to PSD-95 of TARPs, LRRTM2, neuroligin-1 and neuroligin-2 in PSD fractions prepared from six pooled forebrains of *wild type* (WT) mice and six of S*yngap1$^{+/-}$* (HET) mice. The WT animals comprised three 9.5 and two 7.9 week old males and one 12.5 week old female. The HETs comprised three 12.5 week old males, one 7.9 week old male, and two 9.5 week old females. The mean ratio of synGAP to PSD-95 was 25% less in PSDs from the HET mice compared to WT. As we had predicted, the mean ratio of TARPs to PSD-95 showed a small (12%) but significant increase in PSDs from the HET animals compared to WT. We also found a small but significant increase in the mean ratio of LRRTM2 (14%) and neuroligin-2 (9%) to PSD-95. The mean ratio of neuroligin-1 to PSD-95 was unchanged.

Because the number of pooled brains in this previous study was small and WT and HET pools were not perfectly balanced by developmental age or sex, we set out to expand these findings with a larger data set gathered from PSDs isolated from individuals rather than from pooled animals. Data from individuals allowed us to use a more rigorous statistical measure of correlation, the well-established Spearman's rank correlation coefficient r. Comparison of mean levels of two proteins in pooled samples is not a perfect measure of the correlation between the two levels in individuals. It is possible to have a correlation between protein levels in individuals that is not reflected as a difference between mean levels. Spearman's r tests whether a monotonic correlation exists between the rank order of magnitudes of two variables in a data set. We used it to examine the correlation of levels of synGAP with levels of four other proteins in individual PSD fractions. If the rank orders of two variables correlate perfectly, Spearman's r is 1; if there is no correlation, it is zero; and if the ranks are perfectly anti-correlated, it is −1.

When the data were averaged for WT and HET animals in this large data set, we were surprised to find that the mean TARP/PSD-95 ratio in PSDs was not different between WT and HET animals, in contrast to our earlier finding. However, when we calculated Spearman's r for individual data sets, we made the unexpected discovery that a strong inverse correlation between the levels of TARP and synGAP is present only in females and not in males. The large and highly significant inverse correlation in HET females drives a significant inverse correlation data from all HET animals and from all female animals. The inverse correlation is not found in any subset of animals that contains only males. We also established that a similar sex difference is present in rats, as well as mice. We repeated the finding of the earlier study that there is no difference in the level of neuroligin-1 between WT and HET rodents; but, there is a small increase in the amount of neuroligin-2 in HETs. Finally, we made the additional discovery that the level of synGAP correlates positively with the level of the NMDA-type glutamate receptor GluN2B. In other words, the level of NR2B is reduced in the PSDs of HET rodents.

## Results

### Creation of rat synGAP KO by the CRISPR-Cas9 method

CRISPR/Cas9 technology was used to establish a *Syngap1* KO rat line that harbors a frameshift mutation in exon8 of *Syngap1* (*Figure 1A*), which prevents expression of the protein. SynGAP protein expression level is reduced by 50% in HET knockout rats compared to wild-type (WT) and is absent

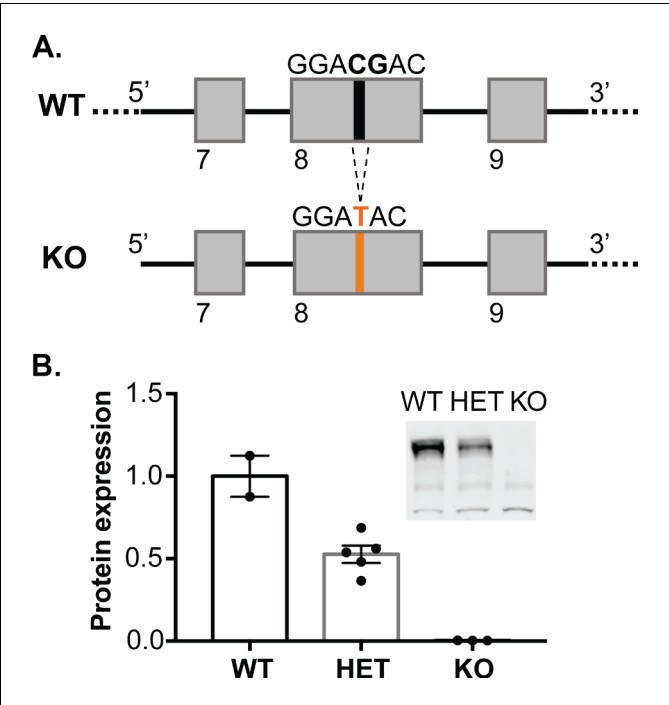

**Figure 1.** Generation of synGAP null rats. (**A**) SynGAP targeting strategy introduces a frame shift mutation into exon 8. (**B**) Quantification of synGAP immunoblots (inset) of cortical homogenates was performed as described in Materials and methods.

in homozygous knockouts (*Figure 1B*). While SynGAP KO rats die perinatally, SynGAP HET rats appear healthy and fertile.

## Average synGAP/PSD-95 and TARP/PSD-95 ratios in WT and HET rodents

We devised a method to isolate PSD fractions from individual mice and rats as described under Materials and Materials and methods. PSD fractions were prepared from the forebrains of 165 individual rodents, comprising 82 WT and 83 HETs. The total sample included 81 females (39 WT, 42 HET), and 84 males (43 WT, 41 HET). In each category, approximately half of the animals were rats and half were mice; approximately half were 7.5 weeks old and half were 12.5 weeks old. The ratio of synGAP/PSD-95 and TARP/PSD-95, averaged over all of the rodents, are summarized in the two bars labeled 'All' (*Figure 2A and B*, left). As expected, the synGAP/PSD-95 ratio (*Figure 2A*, left) is reduced by 22% in HET rodents compared to WT (the WT level is indicated by a dotted line). However, the ratio of TARP to PSD-95 (*Figure 2B*, left) is not significantly different, even when the results were averaged for animals grouped by sex, species, and age (*Figure 2A and B*), except for seven wk old female mice in which the ratio of TARPs to PSD-95 was significantly reduced compared to WT. This value may have been influenced by a developmental effect causing lower overall expression of TARPs in 7 week old mice. We also noted more variability in the averaged ratios of TARP to PSD-95 for females (*Figure 2B*, right) compared to males (*Figure 2B*, middle). Taken as a whole, the averaged results do not reproduce our original finding in *Walkup et al. (2016)*.

## Spearman's correlation coefficient reveals that the synGAP/PSD-95 and TARP/PSD-95 ratios are inversely correlated only in females

We examined the correlation between levels of TARP and synGAP among individual rodents in each sample using the more sensitive measure, Spearman's r. We used Spearman's r rather than Pearson's r for these measurements because many of the data sets showed a non-normal distribution. Pearson's r is valid only for normally distributed data.

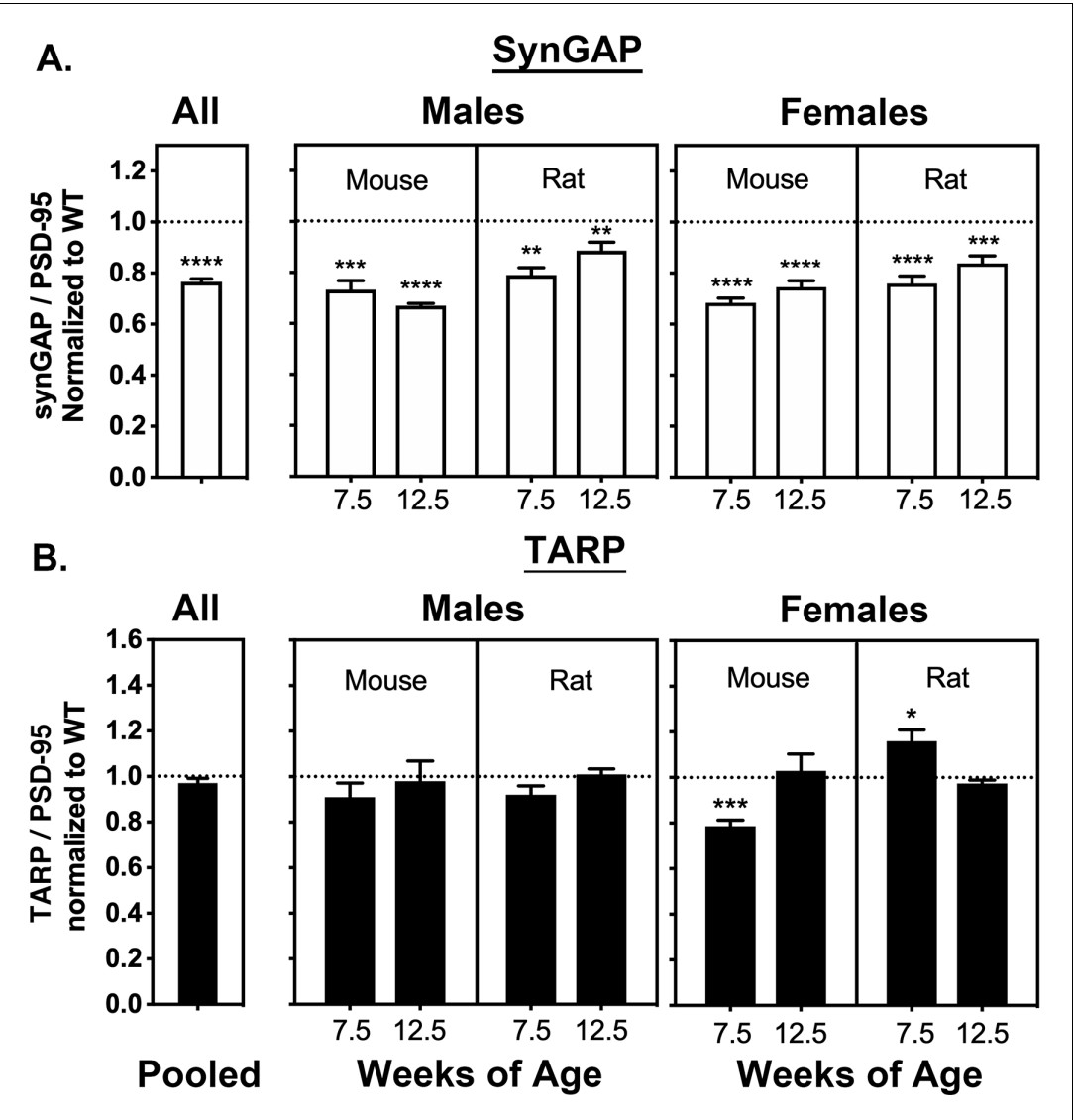

**Figure 2.** Averaged ratios of synGAP and TARPs to PSD-95 in PSDs from WT and HET Rats and Mice. PSDs were purified from the brains of individual animals as described under Materials and methods. The ratios of synGAP to PSD-95 (A) and TARPs to PSD-95 (B) were determined as described under Materials and methods and in *Figure 2—figure supplement 1*. Ratios from HET animals (bars) are normalized to the ratios from WT animals (dotted lines). Antibodies against synGAP, TARPS, and PSD-95 are the same as those used in *Walkup et al. (2016)*. The antibody against synGAP (AB_2287112) recognizes all isoforms of synGAP. The antibody against TARPs (AB_877307) recognizes TARP-γ2, γ3, γ4, and γ8. The sample sizes for each group and the significance tests are as follows. A) all animals WT = 79 and HET = 78, one-tailed Wilcoxon matched-pairs signed rank test; male mouse 7.5 weeks WT = 11 and HET = 9, one-tailed Student T-test; male mouse 12.5 weeks WT = 11 and HET = 8, one-tailed Student T-test with Welch's correction; male rat 7.5 weeks WT = 11 and HET = 10, one-tailed Student T-test; male rat 12.5 weeks WT = 10 and HET = 11, one-tailed Student T-test; female mouse 7.5 weeks WT = 10 and HET = 12, one-tailed Student T-test with Welch's correction; female mouse 12.5 WT = 9 and HET = 9, one-tailed Student T-test; female rat 7.5 weeks WT = 10 and HET = 10, one-tailed Student T-test; female rat 12.5 weeks WT = 9 and HET = 10, one-tailed Student T-test. B) all animals WT = 77 and HET = 80, two-tailed Wilcoxon matched-pairs signed rank test; male mouse 7.5 weeks WT = 10 and HET = 9, two-tailed Student T-test; male mouse 12.5 weeks WT = 10 and HET = 10, two-tailed Mann Whitney test; male rat 7.5 weeks WT = 10 and HET = 10, two-tailed Student T-test; male rat 12.5 weeks WT = 10 and HET = 1, two-tailed Student T-test; female mouse 7.5 weeks WT = 9 and HET = 10, two-tailed Student T-test; female mouse 12.5 WT = 9 and HET = 10, two-tailed Mann Whitney test; female rat 7.5 weeks WT = 10 and HET = 10, two-tailed Student T-test; female rat 12.5

*Figure 2 continued on next page*

*Figure 2 continued*

weeks WT = 9 and HET = 10, two-tailed Student T-test with Welch's correction. Significance: * for p≤0.05, ** for p≤0.01, *** for p≤0.001, and **** for p≤0.0001.

The online version of this article includes the following figure supplement(s) for figure 2:

**Figure supplement 1.** Measurement of densities and calculation of ratios.

The average intensities of staining for proteins differed significantly between the cohorts, presumably because of developmental changes in protein expression. We therefore normalized the ratios for all cohorts to account for these average differences, as described under Materials and methods. The normalization enabled us to look for correlations between ratios among individuals across cohorts.

*Figure 3B and C* contain plots for all WT and all HET animals, respectively. These data show that, at steady state in vivo, lower amounts of synGAP in PSDs from the HET animals (*Figure 3C*) correlate with higher amounts of TARP; whereas there is no correlation in WT animals (*Figure 3B*). This finding supports our original report (*Walkup et al., 2016*).

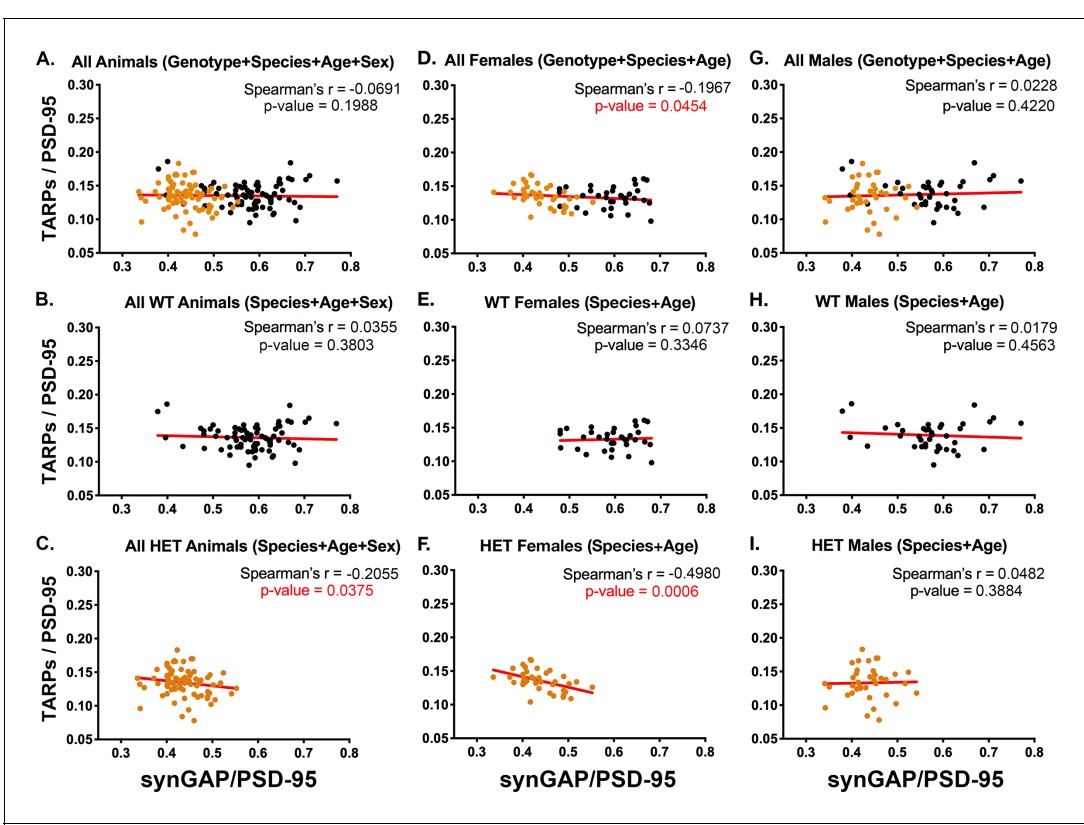

**Figure 3.** Correlation of the ratios TARPs/PSD-95 and synGAP/PSD-95 among individual animals. Each point represents mean ratios for a single animal. Corrected ratios and Spearman's rank correlation coefficients were determined as described under Materials and methods. (A) All animals, including all genotypes, ages, species, and sexes; n = 152. (B) All WT animals, including all ages, species, and sexes; n = 76. (C) All HET animals, including all ages, species, and sexes; n = 76. (D) All female animals, including all genotypes, ages, and species; n = 75. (E) All WT females including all ages and species; n = 36. (F) All HET females, including all ages and species; n = 39. (G) All male animals, including all genotypes, ages, and species; n = 77. (H) All WT males, including all ages and species; n = 40. (I) All HET males, including all ages and species; n = 37. Black symbols, WT; Orange symbols, HET. P-values for Spearman's rank correlation coefficient are one-tailed. Significant p-values are shown in red.

The online version of this article includes the following figure supplement(s) for figure 3:

**Figure supplement 1.** Intensity of PSD-95 bands on immunoblots.

**Figure supplement 2.** Correlation analysis between synGAP/PSD-95 and TARP/PSD-95 for data from 7 and 12 week old female mice and rats.

**Figure supplement 3.** Correlation analysis between synGAP/PSD-95 and TARP/PSD-95 for data from 7 and 12 week old male mice and rats.

The data from all females (*Figure 3D*) show an inverse correlation between the two ratios that just reaches statistical significance. In contrast, the data from all males (*Figure 3G*) shows no correlation. Similarly, WT females and WT males (*Figure 3E and H*) show no correlation. Strikingly, the data from HET females (*Figure 3F*) show the largest inverse correlation of all the data sets, with a Spearman's r = −0.498 and a p-value=0.0006. HET males (*Figure 3I*) show no significant correlation. The strong inverse correlation between the amount of synGAP and the amount of TARP in PSDs from HET females (*Figure 3F*) likely drives the inverse correlation observed for pooled HET animals (*Figure 3C*) and pooled females (*Figure 3D*).

We established that the amounts of PSD-95 per PSD protein are not statistically different between HETs and WT or between male and female subgroups; they are also not correlated with synGAP levels among individuals (*Figure 3—figure supplement 1*). Thus, differences among individuals in the target protein/PSD-95 ratio can be interpreted as actual differences in the concentrations of the target protein in the PSD fractions.

These results mean that, between 7.5 and 12.5 weeks of age, synGAP haploinsufficiency has a much greater effect on the content of TARPs in the PSDs of female animals than in those of males. The simplest explanation is that in HET females, the structure of the PSD, which is determined by multiple equilibria among several proteins, is such that TARP and synGAP compete directly for binding to PSD-95; whereas in HET males, this particular competition is not significant. Possible underlying mechanisms are outlined in the Discussion.

We also compared data sets from mice and rats at 7.5 weeks and 12.5 weeks (*Figure 3—figure supplements 2 and 3*). These data sets were small (9 or 10 animals). Nevertheless, they show a statistically significant inverse correlation between TARP/PSD-95 and synGAP/PSD-95 in HET female mice at both 7.5 and 12.5 weeks. In data from HET rats at 7.5 weeks, the inverse correlation is very close to significance; at 12.5 weeks, it is less significant, but still shows a trend. In the corresponding males, none of the data sets shows a statistically significant inverse correlation. More data would be required to make a definitive conclusion, but the results suggest that competition between synGAP and TARP for binding to PSD-95 in females is more prominent at 7 weeks than at 12 weeks, and more prominent in mice than in rats.

## Effect of synGAP haploinsufficiency on the relative levels of other PSD proteins

In our previous paper, we examined the levels of neuroligins 1 and 2 (NLG-1,–2), and of the surface protein LRRTM2. In this study, we re-examined the effect of reduction of synGAP on the levels of NLG-1 and 2 in the PSD and looked at the effect on levels of GluN2B, a subunit of the NMDA-type glutamate receptor that binds most avidly to PDZ2 of PSD-95. We predicted that the level of GluN2B would be less affected than TARPs or NLGs by reduction of synGAP because synGAP has lower affinity for PDZ2 than for PDZ1 and PDZ3 (*Walkup et al., 2016*).

The ratios of the three proteins to PSD-95 in HET and WT rodents, averaged over the same PSD fractions shown in *Figure 2A and B*, are shown in the bars labeled 'All' in *Figure 4A,B and C* (left). GluN2B exhibits a highly significant reduction of about 10% in HETs compared to WT; NLG-1 shows no change; and NLG-2 increases significantly by about 7%. There is no significant difference in these ratios between rat and mouse, males and females, or between 7.5 week and 12.5 week old animals. The absence of any change in NLG-1 and the slight increase in NLG-2 recapitulate our findings in *Walkup et al. (2016)*.

## Spearman's correlation coefficient reveals that the levels of GluN2B and levels of synGAP in PSDs are positively correlated

*Figure 5* contains ratios of GluN2B to PSD-95 plotted against ratios of synGAP to PSD-95 measured in the same set of individual PSDs shown in *Figure 3A,D and G*. Data for all animals (*Figure 5A*), female animals (*Figure 5B*), and male animals (*Figure 5C*) show positive Spearman's r values of ~0.4 with p-values indicating a highly significant difference from zero. The positive correlation is present in both WT (black) and HET (orange) animals. These results support our hypothesis that synGAP does not compete with GluN2B for binding to PSD-95. To the contrary, the data suggest that a higher level of synGAP leads to higher localization of GluN2B; and, therefore, NMDA-type glutamate receptors, to the PSD (see Discussion).

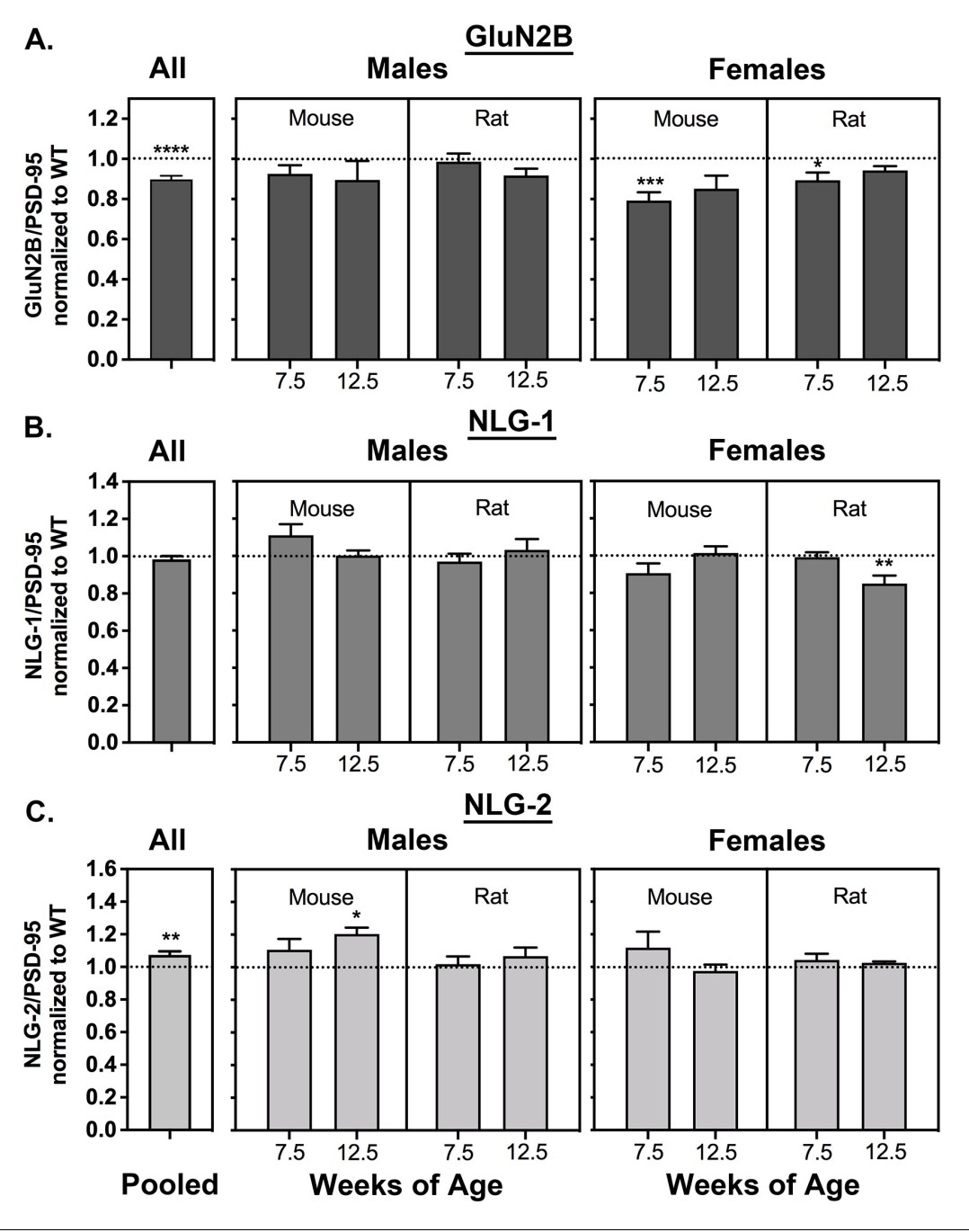

**Figure 4.** Averaged ratios of GluN2B, NLG1, and NLG-2 to PSD-95 in mice and rat syngap1 HETs. PSDs were purified as in *Figure 2*. Ratios were determined as described under Materials and methods. Ratios from HET animals (bars) are normalized to the ratios from WT animals (dotted lines). (**A**) GluN2B/PSD-95. Sample sizes and significance tests are as follows: all animals WT = 81 and HET = 82, two-tailed Wilcoxon matched-pairs signed rank test; male mouse 7.5 weeks WT = 11 and HET = 10, two-tailed Student T-test; male mouse 12.5 weeks WT = 11 and HET = 10, two-tailed Student T-test; male rat 7.5 weeks WT = 11 and HET = 10, two-tailed Student T-test; male rat 12.5 weeks WT = 10 and HET = 11, two-tailed Student T-test; female mouse 7.5 weeks WT = 10 and HET = 12, two-tailed Student T-test; female mouse 12.5 WT = 9 and HET = 9, one-tailed Student T-test; female rat 7.5 weeks WT = 9 and HET = 10, two-tailed Student T-test with Welch's correction; female rat 12.5 weeks WT = 9 and HET = 9, two-tailed Student T-test with Welch's correction. (**B**) NLG-1/PSD-95. Sample sizes and significance tests are as follows: all animals WT = 81 and HET = 83, two-tailed Wilcoxon matched-pairs signed rank test; male mouse 7.5 weeks WT = 10 and HET = 10, two-tailed Student T-test; male mouse 12.5 weeks

*Figure 4 continued on next page*

*Figure 4 continued*

WT = 11 and HET = 10, two-tailed Student T-test; male rat 7.5 weeks WT = 11 and HET = 10, two-tailed Student T-test; male rat 12.5 weeks WT = 10 and HET = 11, two-tailed Student T-test; female mouse 7.5 weeks WT = 10 and HET = 12, two-tailed Student T-test; female mouse 12.5 WT = 10 and HET = 10, two-tailed Student T-test; female rat 7.5 weeks WT = 10 and HET = 11, two-tailed Mann-Whitney test; female rat 12.5 weeks WT = 9 and HET = 10, two-tailed Student T-test with Welch's correction. (**C**) NLG-2/PSD-95. Sample sizes and significance tests are as follows: all animals WT = 79 and HET = 79, one-tailed Wilcoxon matched-pairs signed rank test; male mouse 7.5 weeks WT = 10 and HET = 10, one-tailed Student T-test; male mouse 12.5 weeks WT = 11 and HET = 10, one-tailed Student T-test; male rat 7.5 weeks WT = 11 and HET = 10, one-tailed Student T-test; male rat 12.5 weeks WT = 10 and HET = 11, one-tailed Student T-test; female mouse 7.5 weeks WT = 9 and HET = 12, one-tailed Student T-test with Welch's correction; female mouse 12.5 WT = 10 and HET = 9, one-tailed Student T-test; female rat 7.5 weeks WT = 9 and HET = 10, one-tailed Student T-test; female rat 12.5 weeks WT = 9 and HET = 8, one-tailed Student T-test with Welch's correction. Significance: * for $p \leq 0.05$, ** for $p \leq 0.01$, *** for $p \leq 0.001$, and **** for $p \leq 0.0001$.

## Spearman's correlation coefficient shows no significant correlation between levels of NLG-1 and 2 and levels of synGAP in PSDs

Our previous results showed only a small effect of synGAP haploinsufficiency on the level of NLG-2 in PSDs. The pooled data in *Figure 4C* reproduces those findings. However, the significance of Spearman's r between levels of NLG-2 and synGAP in PSDs shows only a strong trend toward an inverse correlation (*Figure 6A*). As in our previous paper, both *Figure 4B* and *Figure 6B* show no effect of synGAP haploinsufficiency on the amount of NLG-1 in PSDs.

## Discussion

The most striking new result of this research advance is the discovery of a sex difference in the adaptation of the PSD scaffold to synGAP haploinsufficiency. Specifically, we show that a decrease in the steady-state concentration of synGAP in rodent PSDs correlates with a higher concentration of TARPs in PSDs only in females and not in males. In female HETs, the rank correlation coefficient between the concentrations of TARP and synGAP in PSDs is −0.5, which suggests a relatively high competition between the two proteins for binding to PDZ domains of PSD-95 in vivo. This competition does not affect the composition of PSDs in male HETs. SynGAP haploinsufficiency causes about 4% of cases of sporadic intellectual disability (ID) in humans, often accompanied by seizures and autistic behaviors (*Berryer et al., 2013*). If the sex difference in the interaction of synGAP with TARPs in the rodent PSD is also present in humans, it might result in differences between girls and boys in the prevalence of some of the associated symptoms.

The PSD is formed by multiple interactions among the major scaffold proteins and 'client' signaling proteins which are concentrated in the PSD by their association with the scaffold proteins (*Kennedy, 2016*; *Kennedy et al., 2005*; *Sheng and Kim, 2011*). At any one time, the composition of a PSD is a dynamic equilibrium among all the possible protein associations, driven by the relative concentrations of each protein in a spine, and the relative affinities of their mutual binding domains (e.g. *Gray et al., 2006*). The simplest interpretation of our present result is that there is a difference between males and females in the composition or regulation of proteins in PSDs which causes the concentration of TARPs in PSDs to be sensitive to the steady-state concentration of synGAP in females, but not males. This occurs despite the fact that there is no difference between males and females in the amount of synGAP in PSDs of either genotype.

TARPs are subunits of AMPARs, and have been shown to immobilize AMPARs at the synaptic site by binding to PDZ domains of PSD-95 (*Opazo and Choquet, 2011*; *Tomita et al., 2005*). In our recent eLife paper, we postulated that synGAP-α1, which contains a PDZ ligand, competes with TARPs for binding to PSD-95 and therefore helps to limit the number of AMPARs immobilized at the synapse (*Walkup et al., 2016*). This hypothesis has two possible corollaries. One is that transient phosphorylation of synGAP by CaMKII during induction of LTP, which reduces the affinity of its PDZ ligand for PSD-95, will allow more binding of TARP to the PDZ domains; and thus contribute to increased trapping of AMPARs. The second corollary, which is addressed in this study, concerns the steady-state composition of PSDs in *Syngap*[+/-] rodents. If, at steady-state, the concentration of

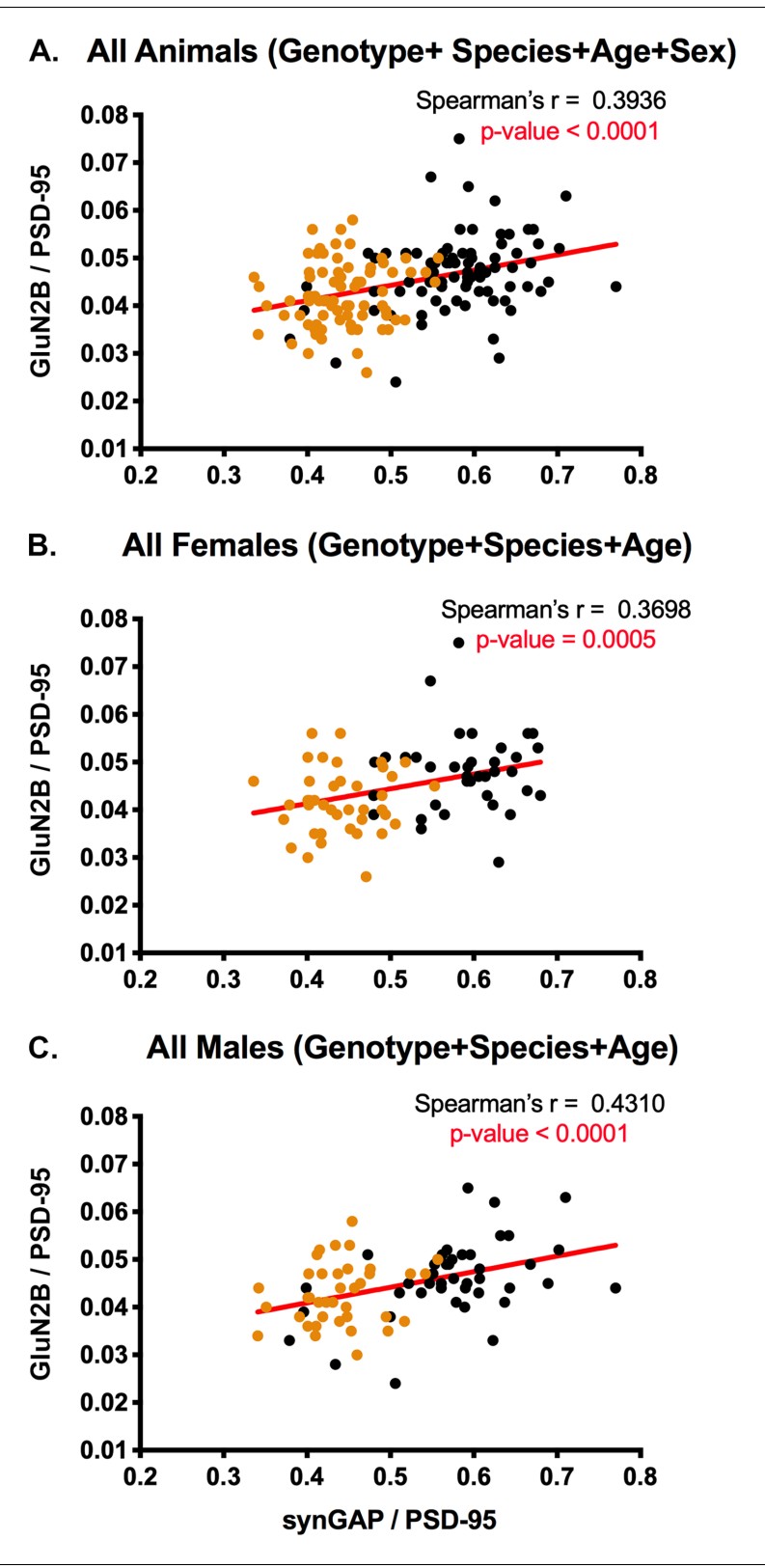

**Figure 5.** Correlation of the ratios GluN2B/PSD-95 and synGAP/PSD-95 in individual animals. Each point represents a single animal. Black, WT; Orange, HET. (**A**) All animals including all genotypes, ages, species, and sexes. n = 158. (**B**) All female animals, including all genotypes, ages, and species. n = 77. C) All male animals, *Figure 5 continued on next page*

*Figure 5 continued*

including all genotypes, ages, and species. n = 81. P-values for Spearman's rank correlation coefficient are one-tailed. Significant p-values are shown in red.

synGAP in WTs is high enough to effectively compete with binding of TARPs to PSD-95, then, in HET rodents, which have a reduced concentration of synGAP, the concentration of TARPs (and thus AMPARs) in the PSD will be higher than in WT. Indeed, we previously reported a higher concentration of TARPs in PSDs isolated from a pool of six HET mice compared to a pool of six WT mice. The pools were only approximately matched for age and sex; the HET pool contained two females, whereas the WT pool contained one (*Walkup et al., 2016*). Because the increase in concentration of TARPs in the HET pool was significant but small, we re- examined the correlation of TARP and synGAP concentrations in a large set of individual PSDs. To do this, we developed a method for isolating PSDs from individual rodents. The analysis supports the hypothesis that synGAP-α1 does indeed compete with steady-state binding of TARPs to PSD-95 in some circumstances.

There are many possible mechanistic explanations for the sex difference in sensitivity of TARPs to the concentration of synGAP. One simple one is that additional protein(s) are present in females that compete with synGAP for binding to PDZ domains of PSD-95. The resulting 'crowding' could make binding of TARP to PSD-95 more sensitive to reduction of synGAP in females.

Another is that an additional protein which can compete with TARPs more strongly than synGAP for binding to PSD-95 is present in male PSDs, but not in female PSDs. In this case, reduction of synGAP in PSDs of males would be expected to have little effect on the concentration of TARPs. Interestingly, despite any differences between males and females in mechanism, the mean steady-state ratio of TARPs to PSD-95 in the two sexes is not significantly different among WTs or HETs.

Two recent studies have documented sex differences in regulation of synaptic plasticity. *Wang et al. (2018)* found that 7 to 12 week old female rodents (the same age range used in our study) have higher synaptic levels of membrane estrogen receptor alpha (mERalpha) and, in contrast to males, require activation of mERalpha for activation of some of the signaling kinases that support long-term potentiation. This difference results in a higher threshold in females for LTP and for some forms of spatial learning. *Jain et al. (2019)*, studying hippocampal slices from 7 to 10 week old rats, found that estrogen-induced plasticity, which occurs in both males and females, requires synergistic activation of L-type $Ca^{2+}$ channels and internal $Ca^{2+}$ stores in females; whereas in males, either of the two sources is sufficient. In addition, activity-dependent LTP requires activation of protein kinase A in females, but not in males. They conclude that there are latent sex differences in mechanisms of synaptic potentiation in which distinct molecular signaling pathways converge to common functional endpoints in males and females. Neither of these studies provides an immediate explanation for our findings, but they support the idea that there are a number of biochemical and structural differences between males and females in the synaptic regulatory apparatus of 7 to 12 week old rodents.

Because the cytosolic tail of the GluN2B subunit binds to the first and second PDZ domains of PSD-95 (*Kornau et al., 1995*), we tested whether the concentration of GluN2B in PSDs is altered by synGAP haploinsufficiency. In contrast to TARPs, the concentration of GluN2B in PSDs shows a strong positive correlation with the amount of synGAP (rank correlation coefficient ≈ 0.4), and is reduced in *Syngap1* HETs. The rank correlation among individual animals is significant in both WT and HET animals and does not differ between males and females (*Figure 5*). This data shows that synGAP plays a role in localizing GluN2B to the PSD, but that it is not the only protein involved.

We reproduced our original finding that the amount of synGAP in the PSD does not influence the amount of NLG-1 and shows a trend toward a small inverse correlation with the amount of NLG-2 in PSDs (*Walkup et al., 2016*). Thus, although both synGAP and NLG-1 (*Irie et al., 1997*) bind to PDZ3 of PSD-95, reduction of synGAP does not increase the steady-state amount of NLG-1 in the PSD in vivo. The simplest explanation is that NLG's have a higher affinity for PDZ3 than synGAP such that synGAP does not compete effectively with them for binding to PDZ3 in vivo. The affinities of NLG-1 and of synGAP for PSD-95 have been estimated by Biacore surface plasmon resonance technology; but, the measurement parameters were not directly comparable. A protein fragment containing all three PDZ domains of PSD-95 was found to have a $K_D$ of ~200 nM for a 16mer with the sequence of the carboxyl terminus of NLG-1 (*Irie et al., 1997*). In our recent study, we found that nearly full length synGAP, missing only the first 100 residues of the amino-terminus, has an affinity for PDZ3

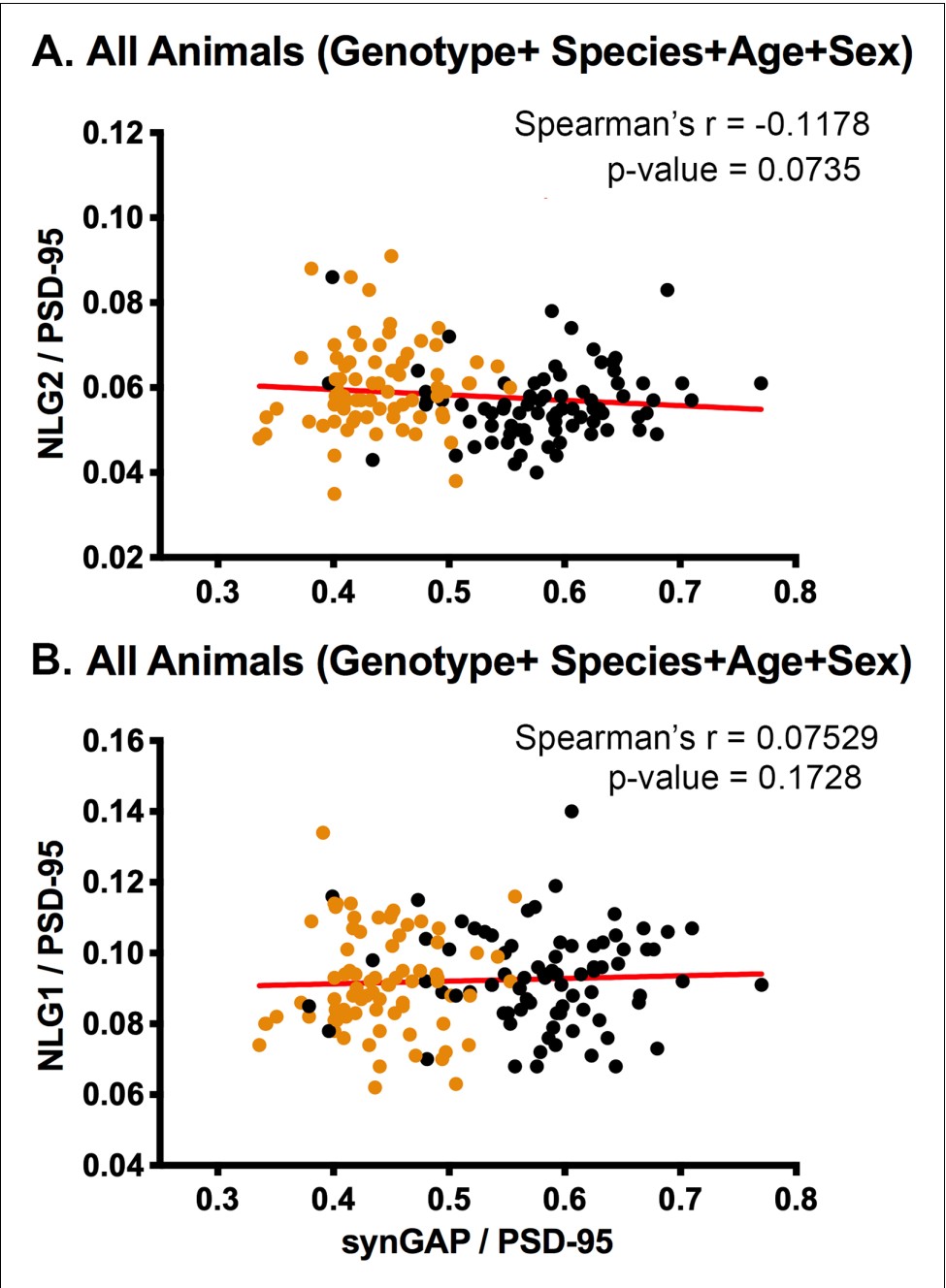

**Figure 6.** Correlation of the ratios of NLG-2/PSD-95 or NLG-1/PSD-95 and synGAP/PSD-95 for individual animals. Each point represents a single animal. Black, WT; Orange, HET. (A) Correlation of NLG-2/PSD-95. All animals, including all genotypes, ages, species, and sexes. n = 153. (B) Correlation of NLG-1/PSD-95. All animals, including all genotypes, ages, species, and sexes. n = 159. P-values of Spearman's rank correlation coefficient indicate no significant correlation.

of ~650 nM and for a fragment containing all three PDZ domains of 5 nM. It is likely that the competition for binding to PDZ3 between NLG's and synGAP is influenced by additional binding sites on PSD-95 or on other scaffold proteins for NLG's and/or synGAP.

These results illustrate the complex role that synGAP plays in modulating the steady-state composition of the PSD. It can compete with some proteins for localization in the PSD and it can help to concentrate others. The results are consistent with the concept that the structure of the PSD is a dynamic equilibrium governed by multiple protein associations and driven by the relative

concentrations of each protein and the affinities of their mutual binding domains. This study has not tested the effect of the concentration of synGAP on the acute re-organization of the PSD that occurs, for example, upon induction of LTP. Our earlier finding that phosphorylation of synGAP by CaMKII, which is activated upon induction of LTP, reduces the affinity of synGAP for all three of the PDZ domains of PSD-95 suggests that post-translational modulation of synGAP's affinity for PSD-95 may initiate transient reorganization of the PSD in stimulated synapses. Disruption of this transient role likely contributes to the phenotypes of synGAP haploinsufficiency, but in ways that may not differ between females and males.

# Materials and methods

## Key resources table

| Reagent type (species) or resource | Designation | Source or reference | Identifiers | Additional information |
|---|---|---|---|---|
| Strain, strain background (*Mus musculus*) | *Syngap1* KO mouse | **Vazquez et al., 2004** | | C57BL/6 |
| Strain, strain background (*Rattus norvegicus*) | *Syngap1* KO rat; *LE-Syngap1^{em1/PWC}* | SAGE Labs, Sigma-Aldrich | *Syngap1^{em1/PWC}* | CRISPR/Cas9 based genome targeting exon 8 |
| Antibody | anti-PSD-95 (Mouse monoclonal) | Caltech Monoclonal Antibody Facility | 50% ammonium sulfate cut of Ascites fluid of 7E3-1B8; AB_212825 | WB (1:10,000) |
| Antibody | anti-synGAP (Rabbit polyclonal) | Pierce | PA1-046; AB_2287112 | WB (1:3,500) |
| Antibody | anti-TARP (Rabbit polyclonal) | EDM Millipore | Ab9876; AB_877307 | WB (1:300) |
| Antibody | anti-GluN2B (Rabbit polyclonal) | Raised in our lab. **Zhou et al., 2007**, see **Figure 2—figure supplement 1** | | WB (1:1000) |
| Antibody | anti-NLG-1 (Rabbit polyclonal) | Synaptic Systems | 129013; AB_2151646 | WB (1:2000) |
| Antibody | anti-NLG-2 (Rabbit polyclonal) | Synaptic Systems | 129202; AB_2151646 | WB (1:1000) |
| Antibody | Alexa Fluor 680 goat anti-mouse IgG | Thermo Fisher Scientific | A28183; AB_2536167 | WB (1 µg/ml) |
| Antibody | IRdye800-conjugated goat-anti-rabbit IgG | Rockland Immunochemicals, Limerick, PA | 611-145-122; AB_1057618 | WB (1 µg/ml) |
| Commercial assay or kit | BCA Protein Assay Kit | Pierce | 23227 | |
| Software, algorithm | Image Studio Light | LI-COR Biosciences | | |
| Software, algorithm | Microsoft Excel | Microsoft | | |
| Software, algorithm | Prism 8 | GraphPad Software, San Diego | | |

## Animals

*Syngap1* KO mice were generated in the Kennedy lab, bred in the Caltech animal facility, and genotyped by polymerase chain reaction as described (**Vazquez et al., 2004**). The rat model LE-*Syngap1^{em1/PWC}*, hereafter referred to as *Syngap1* KO, was produced by SAGE Labs, Sigma-Aldrich (now a subsidiary of Horizon Discovery, Saint Louis, MO, USA) using CRISPR/Cas9 based genome targeting strategies (**Li et al., 2013**). Briefly, CRISPR/Cas9 reagents and sgRNA targeting exon 8 (gtgcatagagcatgtcgtccAGG) were microinjected into pronuclei of fertilized one-cell embryos from Long Evans rats and transplanted into pseudo-pregnant females. Live born pups were genotyped by

Sanger sequencing of exon8 PCR amplicons. Founder #23 displayed a 2 bp deletion 1 bp insertion and was crossed to a wild type Long Evans rat to generate a cohort of heterozygous animals for breeding purposes. Tissue biopsies from pups were used for TaqMan genotyping using two primers (Forward: CCAAGAAGCGATATTACTGCGAGTT and Reverse: GGAAGTGGTCCGTGCATAGA) and two reporter probes (WT: CCTGGACGACATGC and KO: TGCCTGGATACATGC). Homozygous *Syn-GAP* KO rats die perinatally, but heterozygous (HET) *Syngap1* KO rats appear healthy and are fertile. To verify knockout of expression of synGAP, forebrain homogenates of P5 littermate rats (n KO = 3, n HET = 5, n WT = 2) were prepared in radioimmunoprecipitation assay buffer containing protease and phosphatase inhibitors (cOmplete EDTA-free), immunoblotted with a primary antibody raised to panSynGAP (1:4000; Abcam AB77235), and imaged on an Odyssey infrared imaging system (Li-COR Bioscience) as previously described in *Till et al. (2012)*. Total protein levels on the blot were visualised with the Pierce Reversible Protein Stain Kit (Thermo Fisher Scientific, #24580) and quantified with ImageJ gel analyzer software. Protein bands of interest were quantified with Image Studio Lite v5.0 (Li-COR Bioscience). The expression level of each protein of interest was first normalized to total protein, followed by normalization of the data to the average wild-type levels, which were considered to be 100%.

Upon weaning each animal was given an ear punch ID associated with a unique ID number, sexed, and genotyped. Tissue from the ear punch was used for genotyping both mice and rats, although some mice were genotyped with tissue from the tail. Mice were genotyped by polymerase chain reaction as described (*Vazquez et al., 2004*) and rats were genotyped via sequencing performed by Transnetyx, Inc Cordova, TN. The ID numbers linked the genotype and all other metadata for an animal to its PSD samples.

## Preparation of PSD fractions

PSD fractions were prepared from individual WT and HET mice and rats that were either 7.5 weeks or 12.5 weeks old, by a modification of a standard method (*Cho et al., 1992*; *Cohen et al., 1977*). ID numbers were assigned randomly with respect to genotype, and were used to label and track tissue samples and extracts after harvesting from the animal. Animals were killed by decapitation according to a protocol approved by the Caltech Institutional Animal Care and Use Committee. The following steps were carried out at 4° C. Forebrains were dissected from each animal, rinsed in Buffer A (0.32 M sucrose, 1 mM NaHCO$_3$, 1 mM MgCl$_2$, 0.5 mM CaCl$_2$, 0.1 mM phenylmethylsulfonyl chloride [PMSF, Sigma Millipore, St. Louis, MO]). Each individual forebrain was homogenized in Buffer A (10% w/v, 4.5 ml for mice and 13.5 ml for rats) with 12 up and down strokes of a teflon/glass homogenizer at 900 rpm. Homogenates were subjected to centrifugation at 1400 × g for 10 min. The pellet was resuspended in Buffer A to 10% w/v (3.8 ml for mice, 12 ml for rats), homogenized (three strokes at 900 rpm) and subjected to centrifugation at 710 g for 10 min. The two supernatants were combined and subjected to centrifugation at 13,800 g for 10 min. The resulting pellet was resuspended in Buffer B (0.32 M sucrose, 1 mM NaHCO$_3$; 2 ml for mice, 8 ml for rats), homogenized with 6 strokes at 900 rpm in a teflon/glass homogenizer, and layered onto a discontinuous sucrose gradient (equal parts 0.85 M, 1.0 M, and 1.2 M sucrose in 1 mM NaH$_2$CO$_3$ buffer; 10.5 ml for mice, 30 ml for rats). Gradients were subjected to centrifugation for 2 hr at 82,500 g in a swinging bucket rotor. The synaptosome-enriched layer at the interface of 1.0 and 1.2 M sucrose was collected, diluted with Buffer B, 7 ml for mice and 20 ml for rats, then added to an equal volume of Buffer B containing 1% Triton (10% X-100 Surfact-Amps, Thermo Fisher Scientific, Waltham, MA). The mixture was stirred for 15 min at 4°C and subjected to centrifugation for 45 min at 36,800 g. The pellet, which contained the PSD-enriched Triton-insoluble fraction, was resuspended in 0.5–1.0 ml 40 mM Tris pH 8 with a 21-gauge needle attached to a 1 ml syringe, and then homogenized with six strokes at 900 rpm in a teflon/glass homogenizer. Samples were flash-frozen, and stored at −80° C. Protein concentrations were determined by the bicinconic acid method (Thermo Fisher Scientific). Individual mouse brains yielded ~1.25 mgs and individual rat brains yielded ~3.5 mgs protein. The yield was sufficient to measure the ratio of amounts of five separate proteins to PSD-95 for each individual animal.

## Immunoblots

To measure each of the five proteins, an equal amount of protein from each sample (5 µg) was dissolved in SDS-PAGE sample buffer (50 mM Tris-HCL pH 6.8, 2% SDS, 10% glycerol, 5% β-

mercaptoethanol, 0.005% bromophenol blue), heated at 90℃ for 5 min, loaded into acrylamide gels (8% gel for analysis of synGAP, GluN2B, NLG-1 and 2; or 12% for TARPs), fractionated, and electrically transferred to PVDF membranes in 25 mM Tris, 150 mM glycine at 250V for 2–3 hr at 4 ℃. All gels contained a mixture of males and females randomized by genotype. Completeness of transfer was checked by staining of gels with Gel Code Blue (Thermo Fisher Scientific) after transfer. Membranes were washed for 5 mins three times in 20 mM Tris, pH 7.6, 150 mM NaCl (TBS), blocked with Odyssey blocking buffer (LI-COR Biosciences, Lincoln NE) for 45 mins at RT, washed for 5 mins three times in TBS plus 0.1% tween (TBST) and then incubated in primary antibodies dissolved in TBST plus 5% BSA ON at 4 ℃. Each blot was incubated with a mixture of mouse anti-PSD-95 (50% ammonium sulfate cut of Ascites fluid of 7E3-1B8, AB_212825, dilution 1:10,000), and one of the following; rabbit anti-synGAP (Pierce PA1-046, AB_2287112 dilution 1:3500), rabbit anti-TARP (EDM Millipore Ab9876, AB_877307 dilution 1:300), rabbit anti-GluN2B (raised in our lab, *Zhou et al., 2007*, see *Figure 2—figure supplement 1*, dilution 1:1000), rabbit anti-NLG-1 (Synaptic Systems 129013 AB_2151646 dilution 1:2000), or rabbit anti-NLG-2 (Synaptic System 129202 dilution 1:1000). Membranes were washed for 5 min three times in TBST and incubated for 45 min at RT in 5% nonfat milk in TBST with a mixture of secondary antibodies including Alexa Fluor 680 goat anti-mouse IgG (Thermo Fisher Scientific A28183, AB_2536167, 1 µg/ml) to label PSD-95, and IRdye800-conjugated goat-anti-rabbit IgG (Rockland Immunochemicals, Limerick, PA, 611-145-122, AB_1057618, 1 µg/ml) to label the target proteins. Membranes were then given three 5 min washes in TBST, followed by three 5 min washes in TBS. Bound antibodies were visualized with the Odyssey Infrared Imaging System (LI-COR Biosciences).

## Data quantification and analysis

For each gel lane (representing a single animal) the ratio of the target protein (NLG-1, NLG-2, TARPs, GluN2B, or synGAP) to PSD-95 was calculated (see *Figure 2—figure supplement 1* and *Supplementary file 1*). Regions of Interest (ROI) were drawn around each protein band and the intensity within the box was determined with the use of Image Studio Light software supplied by LI-COR. To measure background, a box of the same size was placed in the lane in an unstained region above or below the band of interest. The fluorescence intensity values were transferred to Microsoft Excel. For both the target protein and PSD-95, background was subtracted from the signal, then the ratio of the two background-corrected signals was calculated. At least three technical replicates of the synGAP signal were performed for each animal; and, in general, six technical replicates were performed for the other targets.

To gather the data, cohorts of 7.5 week old mice, 7.5 week old rats, 12.5 week old mice, and 12.5 week old rats were processed separately. Because of the large number of animals within each cohort (~40), gels were run and the samples were quantified over a few days and sometimes with different lots of antibodies. To normalize intensity signals within each cohort between different days and different lots of antibodies for *Figures 2* and *4*, we first grouped the ratios for each target (e.g synGAP/PSD-95) according to the genotype and sex of the animals. For each combination of genotype and sex (e.g. WT male), we averaged the ratios determined for each target on each day, then calculated the overall average of these daily averages. To obtain correction factors that compensated for systematic small variations in signals on different days and for different lots of antibodies, we calculated the ratio of the overall average to each daily average. Each ratio for individual animals (ie. from a single lane) was then adjusted by multiplying by the appropriate correction factor. Eighty percent of the correction factors fell between 0.5 and 1.75. One percent were larger than 2.75 and four percent were less than 0.5. After normalization within cohorts, the ratios were used for the analyses in *Figures 2* and *4*. This procedure allowed us to correct for technical variation within cohorts while preserving true differences in ratios based on species, age, genotype, or sex. Analyses for significance were carried out and graphs were created with Prism 8 (GraphPad Software, San Diego). The D'Agostino-Pearson omnibus test was used to determine the normality of the data sets. The means of groups of data were compared for significant differences with one- or two-tailed Mann-Whitney test (when non-normal), or one- or two-tailed paired T-Test, as indicated. If the variances of the two groups were found to be different using an F test, a one- or two-tailed T-Test with a Welch's correction was applied.

To test rigorously among individual animals for correlations between the ratio of synGAP to PSD-95 and the ratio of the other target proteins to PSD-95 (*Figures 3*, *5* and *6*), We corrected for

differences in intensity of signals between the four cohorts (i.e. 7.5 week old mice, 7.5 week old rats, 12.5 week old mice, and 12.5 week old rats) which had been analyzed separately. (The previous normalization only corrected for technical variation within each cohort.) Data for each cohort was divided into ratios from WT males, WT females, HET males, and HET females. Averages of the ratios for each protein within each of these groups, were calculated. The averaged ratios for each group were then further averaged across the cohorts. A normalization factor was calculated for each protein in each cohort by dividing the overall average for all the cohorts by the average for each cohort. Then the appropriate normalization factor was applied to individual data points in each cohort. This sequence corrected for variation in the average intensities of signals between cohorts and groups (for example, the overall lower expression of TARPS in 7.5 week old HET females [*Figure 1B*]) and allowed us to look for correlations between ratios among the individuals across cohorts. To determine the correlations between ratios shown in *Figures 3*, *5* and *6*, we treated data as non-normal and calculated one-tailed Spearman rank correlation coefficients with Prism software.

# Additional information

### Funding

| Funder | Grant reference number | Author |
| --- | --- | --- |
| National Institute of Mental Health | MH15456 | Tara L Mastro<br>Anthony Preza<br>Mary B Kennedy |
| Allen and Lenabelle Davis Foundation | Professorship | Mary B Kennedy |
| National Science Foundation | Postdoctoral Fellowship | Tara L Mastro |
| Department of Biotechnology, Ministry of Science and Technology | | Sumantra Chattarji<br>Peter C Kind |
| Simons Foundation | Grant 529085 | Peter C Kind |
| Patrick Wild Centre | | Sally M Till<br>Peter C Kind |
| Medical Research Council | MR/P006213/1 | Sally M Till<br>Peter C Kind |

The funders had no role in study design, data collection and interpretation, or the decision to submit the work for publication.

### Author contributions

Tara L Mastro, Conceptualization, Data curation, Formal analysis, Supervision, Funding acquisition, Validation, Investigation; Anthony Preza, Data curation, Investigation, Writing - review and editing; Shinjini Basu, Resources, Investigation; Sumantra Chattarji, Peter C Kind, Resources, Funding acquisition; Sally M Till, Resources, Investigation, Visualization; Mary B Kennedy, Resources, Formal analysis, Supervision, Funding acquisition, Validation, Visualization, Project administration

### Author ORCIDs

Tara L Mastro https://orcid.org/0000-0003-1302-9753
Sumantra Chattarji http://orcid.org/0000-0001-9962-3635
Peter C Kind http://orcid.org/0000-0002-4256-9639
Mary B Kennedy https://orcid.org/0000-0003-1369-0525

### Ethics

Animal experimentation: This study was performed in strict accordance with the recommendations in the Guide for the Care and Use of Laboratory Animals of the National Institutes of Health. All of the animals were handled according to approved institutional animal care and use committee (IACUC) protocols (1034-18) of California Institute of Technology.

**Decision letter and Author response**
Decision letter https://doi.org/10.7554/eLife.52656.sa1
Author response https://doi.org/10.7554/eLife.52656.sa2

## Additional files

### Supplementary files

• Supplementary file 1. Example calculations of synGAP/PSD-95 ratio and TARPs/PSD-95 ratio for animals 33 and 34. The Table shows the steps to calculate a synGAP/PSD-95 ratio and a TARPs/PSD-95 ratio from digital data recorded from one pair of gels containing samples from animals 33 and 34 (see *Figure 2—figure supplement 1*). Data were collected from at least three technical replicates of synGAP gel lanes for each animal and usually six technical replicates of target gel lanes. The resulting ratios were averaged.

• Transparent reporting form

### Data availability

All data generated or analysed during this study are included in the manuscript and supporting files.

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
