## [Decision Letter]

Thank you for submitting your article "A sex difference in the composition of the rodent postsynaptic density" for consideration by *eLife*. Your article has been reviewed by two peer reviewers, and the evaluation has been overseen by Leslie Griffith as Reviewing Editor and Eve Marder as the Senior Editor. The reviewers have opted to remain anonymous.

The reviewers have discussed the reviews with one another and the Reviewing Editor has drafted this decision to help you prepare a revised submission.

Summary:

This paper builds on a previous report by the Kennedy lab showing that synGAP may play a role in regulation of PSD composition. In this Advance, the authors develop a single-animal PSD prep and find that, surprisingly, there is a sex difference in the effects of heterozygosity for synGAP. While this is a very interesting finding, there were several issues raised by the reviewers that need to be addressed.

Essential revisions:

1) The authors need to be more transparent about the ratio data so a reader can understand what underlies the means and correlations they plot in figures. Ruling out PSD-95 differences based on means, but ruling in a synGAP/TARP relationship based on correlations that are not evident in means seems potentially problematic.

2) The authors need to relate their new data more clearly to the previous results and conclusions. As an Advance, it should be completely clear how these results are both related to and advance understanding of the relationship between synGAP and TARPs. The primary issue is with Figure 2. Both synGAP/PSD-95 *and* TARPs/PSD-95 go *down* relative to WT in HET female mice at 7.5 weeks; there is no effect on TARPs in 12.5 week female mice, despite the expected decrease in synGAP. Thus, with the exception of 7.5 week HET female rats, in which TARPs go up when synGAP goes down, the relationships shown in the previous paper are not recapitulated. It is also very hard to understand how such a small fraction of females in the first paper could have accounted for those results unless those data sets were seriously underpowered and did not reflect the true underlying distribution of the phenomena.

3) The authors need to show primary data to support the rigor of their analyses e.g. some representative immunoblots.

4) Most of the pairwise statistics are one-tailed. The authors need to do two-sided statistical tests or genuinely justify one-sided tests. It is possible they did have a hypothesis to justify one-sided tests (based on the previous paper). But Figure 2 (in which synGAP and TARPs do not consistently show an opposite regulation) calls this into question.

5) Since there are no sex difference apparent in WT animals, the title of the paper should reflect this and focus on the sex difference in the ability of animals to respond to genetic insult.

---

## [Author Response]

Essential revisions:1) The authors need to be more transparent about the ratio data so a reader can understand what underlies the means and correlations they plot in figures. Ruling out PSD-95 differences based on means, but ruling in a synGAP/TARP relationship based on correlations that are not evident in means seems potentially problematic.

We have addressed concerns 1 and 3 in the following ways:

First, we added Figure 2—figure supplement 1 that shows two examples of blots from which the ratio data was calculated. We include a step by step explanation of how the data was transformed into the indicated ratios. We include Supplementary file 1 that provides two examples of the numbers generated at each step.

The second sentence in concern no. 1 is a fair point. We have now included in Figure 3—figure supplement 1, a correlation plot and determination of Spearman’s r for the relationship between synGAP levels and PSD-95 levels in the data set for 7.5 week old mouse HET females. This graph (Figure 3—figure supplement 1C) shows that the correlation between synGAP/PSD-95 and TARPs/PSD-95 in this data set is not related to a change in PSD-95 levels in the synGAP HETs.

2) The authors need to relate their new data more clearly to the previous results and conclusions. As an Advance, it should be completely clear how these results are both related to and advance understanding of the relationship between synGAP and TARPs. The primary issue is with Figure 2. Both synGAP/PSD-95 and TARPs/PSD-95 go down relative to WT in HET female mice at 7.5 weeks; there is no effect on TARPs in 12.5 week female mice, despite the expected decrease in synGAP. Thus, with the exception of 7.5 week HET female rats, in which TARPs go up when synGAP goes down, the relationships shown in the previous paper are not recapitulated. It is also very hard to understand how such a small fraction of females in the first paper could have accounted for those results unless those data sets were seriously underpowered and did not reflect the true underlying distribution of the phenomena.

The study presented here is directly related to the previous results in the following way.

The original comparison of levels of TARP and synGAP between pools of six wild type animals and six HET animals were made with pools that contained animals between 7.5 and 12.5 weeks old that were not exactly balanced for developmental age or sex. We were, and are, aware that developmental differences in expression of synGAP or TARPs between animals at these ages might have contributed to our findings. Therefore, we undertook to collect larger individual data sets of animals at each age and of each sex in order to more carefully examine the relationship between amounts of synGAP and amounts of TARPs in PSDs. The data set of measurements on individual PSDs enabled us to use the well-established statistical measure, Spearman’s r, a more rigorous measure of correlation between two measurements. It is exactly because of the possibility that the original data might have been underpowered that we undertook the present study to examine the issue more rigorously. We also indicate in the manuscript a related reason for this follow up study. Namely, comparison of mean ratios is not the most accurate or sensitive way to test for a correlation between the levels of two proteins. It is entirely possible to have a situation in which the levels of two proteins are correlated when examining each individual; however, the mean levels of the two proteins, averaged over all individuals, appear uncorrelated. As we explained in the manuscript, Spearman’s r is a well-known statistical test designed to reveal correlations based on measurements from each individual in a group.

We also added a study of rats, for which we have obtained synGAP heterozygote mutants, in order to establish whether mice and rats differ.

In response to the reviewer’s request, we moved to the Introduction (second paragraph) the explanation of the relationship to the original study, and of our decision to gather individual data to permit a more rigorous study.

The study involved the sacrifice of 160 mice and rats. We would have had a hard time justifying such a detailed study without the original finding. Investigators need to publish data at reasonable intervals and cannot always wait until each aspect of a finding is explored to the fullest. Notably, except for the sex difference, the relationships indicated in our original study between TARP and synGAP, neuroligin-1 and synGAP, and neuroligin-2 and synGAP are correct, despite the apparent “underpowering.” That is the data. We were, and are, not in the position of those proposing clinical trials, in which a meaningful estimate of needed “power” can be calculated prior to gathering the actual data.

In summary, we did this follow up study principally to test in a more sensitive way for a correlation between synGAP and TARP, and between synGAP and the other proteins; and also to determine the variability among individuals of levels of synGAP, TARP, neuroligin-1, neuroligin-2, and GluN2B in PSDs.

3) The authors need to show primary data to support the rigor of their analyses e.g. some representative immunoblots.

Please see our response to comment #1.

4) Most of the pairwise statistics are one-tailed. The authors need to do two-sided statistical tests or genuinely justify one-sided tests. It is possible they did have a hypothesis to justify one-sided tests (based on the previous paper). But Figure 2 (in which synGAP and TARPs do not consistently show an opposite regulation) calls this into question.

We have replaced one-tailed T-tests with two-tailed T-tests for the data in Figure 2B. In Figure 3, we kept one-tailed T-tests since our hypothesis still predicted an inverse relationship between synGAP and TARP levels. In Figure 4, we substituted two-tailed T-tests since we did not have a pre-existing hypothesis about this data. We also used a two-tailed T-test in Figure 4B since NLG-1 did not appear to have a consistent relationship to synGAP. In Figure 4C, we hypothesized that NLG-2 would show an anti-correlation with synGAP, so we used a one-tailed T-test. In Figure 5, we hypothesized, based on Figure 4A, that the level of GluN2B would correlate positively with synGAP, therefore, we used one-tailed T-tests. In Figure 6, neither one-tailed nor two-tailed tests indicated significant correlation. Although, the one-tailed T-test in Figure 6A suggested a trend toward anticorrelation.